# Stable Up-Down Quark Matter Nuggets, Quark Star Crusts, and a New Family of White Dwarfs

Lang Wang [1], Jian Hu [1], Cheng-Jun Xia [1,*], Jian-Feng Xu [2,*], Guang-Xiong Peng [3,4,5,*] and Ren-Xin Xu [6,7,*]

1 School of Information Science and Engineering, NingboTech University, Ningbo 315100, China; wanglang@nit.zju.edu.cn (L.W.); 13655826788@139.com (J.H.)
2 School of Physics & Electrical Engineering, AnYang Normal University, AnYang 455000, China
3 School of Nuclear Science and Technology, University of Chinese Academy of Sciences, Beijing 100049, China
4 Theoretical Physics Center for Science Facilities, Institute of High Energy Physics, P.O. Box 918, Beijing 100049, China
5 Synergetic Innovation Center for Quantum Effects and Application, Hunan Normal University, Changsha 410081, China
6 School of Physics, Peking University, Beijing 100871, China
7 Kavli Institute for Astronomy and Astrophysics, Peking University, Beijing 100871, China
* Correspondence: cjxia@nit.zju.edu.cn (C.-J.X.); jfxu@aynu.edu.cn (J.-F.X.); gxpeng@ucas.ac.cn (G.-X.P.); r.x.xu@pku.edu.cn (R.-X.X.)

**Abstract:** The possible existence of stable up-down quark matter (*ud*QM) was recently proposed, and it was shown that the properties of *ud*QM stars are consistent with various pulsar observations. In this work we investigate the stability of *ud*QM nuggets and found at certain size those objects are more stable than others if a large symmetry energy and a small surface tension were adopted. In such cases, a crust made of *ud*QM nuggets exists in quark stars. A new family of white dwarfs comprised entirely of *ud*QM nuggets and electrons were also obtained, where the maximum mass approaches to the Chandrasekhar limit.

**Keywords:** quark matter; up-down quark nuggets; quark star crusts; white dwarfs

## 1. Introduction

At increasing density of baryonic matter, it is expected that a deconfinement phase transition will take place and form quark matter. The properties of quark matter is of particular interest to us since its absolute stability would permit an explanation of dark matter within the framework of the standard model [1]. In the beginning of the 1970s, it was suggested that strange quark matter (SQM) comprised of *u*, *d*, and *s* quarks may be more stable than nuclear matter [1–3], which can exist in various forms, e.g., strangelets [4–7], nuclearites [8,9], meteorlike compact ultradense objects [10], and strange stars [11–13]. Nevertheless, the absolute stability of SQM was challenged by chiral models due to a too large strange quark mass with dynamical chiral symmetry breaking [14,15]. Then SQM only exists in extreme conditions such as the center of compact stars [16–22] and heavy-ion collisions [23,24]. In recent years, an interesting proposition was raised suggesting that quark matter comprised of only *u* and *d* quarks (*ud*QM) may be more stable [25], so that *ud*QM nuggets and *ud*QM stars can exist in the Universe. Due to a much smaller surface tension, the ordinary nuclei would not decay into *ud*QM nuggets [25,26]. In fact, it was shown that in a large parameter space the energy per baryon of *ud*QM nuggets is larger than 930 MeV at $A \lesssim 300$ [25]. The properties of nonstrange quark stars and their astrophysical implications are then examined extensively in recent years, e.g., those in References [27–30]. In particular, the merger of binary quark stars would eject *ud*QM nuggets into space. If those objects become supercritically charged, the $e^+e^-$ pair production would inevitably start and release a large amount of energy. The positron emission of the supercritically charged objects are thus expected to play important roles in the short $\gamma$-ray burst during

the merger of binary quark stars, the 511 keV continuum emission, and the narrow faint emission lines in X-ray spectra from galaxies and galaxy clusters [26].

Based on various investigations, it was shown that the interface effects of quark matter play key roles in the properties of strangelets, $ud$QM nuggets, compact stars, and the processes of quark-hadron transition [26,31,32]. The energy contribution due to the interface effects is often taken into account with a surface tension $\sigma$, while its exact value is still veiled in mystery. Adopting the bag model [33], linear sigma model [34–36], NJL model [37,38], three-flavor Polyakov-quark-meson model [39], Dyson-Schwinger equation approach [40], equivparticle model [41], nucleon-meson model [42], and Fermi gas approximations [43,44], recent estimations indicate a small value with $\sigma \lesssim 30$ MeV/fm$^2$, while larger values were obtained in previous studies [45–47].

In the framework of the bag model, it was shown that a small strangelet can be destabilized substantially if $\sigma^{1/3} \approx B^{1/4}$ with $B$ being the bag constant [4], while the minimum baryon number for metastable strangelets $A_{\min} \propto \sigma^3$ [5,48]. Depending on the values of surface tension, large strangelets and strange stars will face very different fates. On the one hand, if a moderate value for $\sigma$ is adopted, larger strangelets are more stable than smaller ones and strange stars' surfaces are likely bare [49]. On the other hand, if $\sigma$ is smaller than a critical value $\sigma_{\mathrm{crit}}$, large strangelets will decay via fission [50] and strange stars' surfaces may fragment into crystalline crusts [51]. Adopting linearization for the charge density, it was shown that the critical surface tension can be obtained with [50]

$$\sigma_{\mathrm{crit}} = 0.1325 n_Q^2 \lambda_D / \chi_Q, \tag{1}$$

where $n_Q$ is the charge density, $\lambda_D = 1/\sqrt{4\pi\alpha\chi_Q}$ the Debye screening length, and $\chi_Q = \sum_i q_i \frac{\partial n_Q}{\partial \mu_i}$ the electric charge susceptibility of quark matter at zero electric charge chemical potential $\mu_e = 0$. Assuming noninteracting SQM, Equation (1) suggests $\sigma_{\mathrm{crit}} \propto m_s^4$ with $m_s$ being the strange quark mass [51,52]. As we increase $m_s$, the strangeness per baryon $f_s$ for $\beta$-stable SQM decreases and eventually reaches $f_s = 0$, where SQM is converted into $ud$QM. We thus expect that the critical surface tension of $ud$QM is much larger than that of SQM, so it is more likely that there exist $ud$QM nuggets at certain size that are more stable than others. Additionally, varying the symmetry energy of quark matter will alter the values of $n_Q$, $\lambda_D$, $\chi_Q$, and consequently $\sigma_{\mathrm{crit}}$ according to Equation (1). Since it was shown that the symmetry energy of quark matter plays an important role on the structures of quark stars [53–57], in this work we investigate its impact on the properties of small objects such as $ud$QM nuggets.

The purpose of our current study is thus twofold, i.e., investigate the properties of $ud$QM nuggets with various symmetry energies and discuss their implications on $ud$QM stars' structures. The paper is organized as follows. In Section 2, we discuss briefly the equivparticle model and present the corresponding Lagrangian density. To investigate the impact of symmetry energy, an isospin dependent term is added to the quark mass scaling. Then the properties of $ud$QM nuggets are investigated adopting the method discussed in our previous publications [58–61], where the stability window for $ud$QM is obtained according to the binding energy of the heaviest $\beta$-stable nucleus $^{266}$Hs. The properties of $ud$QM stars with and without crusts are then examined in Section 4 according to the stability of $ud$QM nuggets. We draw our conclusion in Section 5.

## 2. Equivparticle Model

As an example, in this work we adopt the equivparticle model to investigate the properties of quark matter and their nuggets. The strong interactions are included with density-dependent quark masses in the equivparticle model, while quarks are considered as quasi-free particles [53,62–71]. It is thus straightforward to write out the Lagrangian density, i.e.,

$$\mathcal{L} = \sum_{i=u,d,s} \bar{\Psi}_i \left[ i\gamma^\mu \partial_\mu - m_i(n_{\mathrm{b}}) \right] \Psi_i, \tag{2}$$

where $\Psi_i$ represents the Dirac spinor of quark flavor $i$, $m_i(n_b)$ the equivalent mass with $n_b$ being the baryon number density. The Coulomb interactions can also be included by adding photon field in the Lagrangian density [41].

The strong interaction among quarks are then reproduced by equivalent quark masses, where many different mass scalings were proposed. For example, for density dependent masses $m_i(n_b) = m_{i0} + m_I(n_b)$ with $m_{u0} = 2.2$ MeV and $m_{d0} = 4.7$ MeV being the current masses of $u$ and $d$ quarks [72], the inversely linear scaling $m_I = B/3n_b$ was obtained by reproducing bag model results in the limit of vanishing densities [73]. Considering the contributions of linear confinement and adopting linearization for the in-medium chiral condensates, an inversely cubic scaling $m_I = Dn_b^{-1/3}$ was derived [74], while the one-gluon-exchange interaction was later included with $m_I = Dn_b^{-1/3} - Cn_b^{1/3}$ [66]. Meanwhile, in the limit of large densities, perturbation theory suggests repulsive interactions among quarks and the quark mass scaling becomes $m_I = Dn_b^{-1/3} + Cn_b^{1/3}$ [68]. An isospin dependent term was also introduced to examine the impacts of quark matter symmetry energy, which was given by $m_I = Dn_b^{-1/3} - \tau_i \delta D_I n_b^{\alpha} e^{-\beta n_b}$ with $\tau_i$ being the third component of isospin for quark flavor $i$ and $\delta = 3(n_d - n_u)/(n_d + n_u)$ the isospin asymmetry [53].

In this work we adopt the following quark mass scaling

$$m_I(n_b) = Dn_b^{-1/3} + Cn_b^{1/3} + C_I \delta^2 n_b. \tag{3}$$

The first term corresponds to linear confinement with the confinement parameter $D$ connected to the string tension $\sigma_0$, the chiral restoration density $\rho^*$, and the sum of the vacuum chiral condensates $\sum_q \langle \bar{q}q \rangle_0$ [74]. The second term represents the contribution of one-gluon-exchange interaction for $C < 0$ [66] and the leading-order perturbative interaction for $C > 0$ [68], where in both cases $C$ is connected to the strong coupling constant $\alpha_s$. Finally, we have added the third term in Equation (3) to account for the quark matter symmetry energy, which is increasing with $C_I$. In fact, in addition to the kinetic contribution, it was shown that the formation of $u$-$d$ quark Cooper pairs (2SC phase) could effectively enhance the quark matter symmetry energy [54]. The impacts of quark matter symmetry energy on compact star properties were extensively investigated in the past few years, e.g., those in References [53–57]. Note that in contrast to the mass scaling proposed in Reference [53], here $m_I$ is identical for different quark flavor, i.e., neglecting the isovector-scalar channel. Meanwhile, as will be discussed later, the third term leads to the isovector contributions in the vector potentials of quarks.

Adopt mean-field and no-sea approximations, the energy density $E$ of quark matter at zero temperature is obtained with

$$E = \sum_i \int_0^{\nu_i} \frac{g_i p^2}{2\pi^2} \sqrt{p^2 + m_i^2} \, dp = \sum_i \frac{g_i m_i^4}{16\pi^2} f\left(\frac{\nu_i}{m_i}\right), \tag{4}$$

where $f(x) = \left[x(2x^2 + 1)\sqrt{x^2 + 1} - \mathrm{arcsh}(x)\right]$, $g_i$ (= 6 for quarks and 2 for leptons) the degeneracy factor for particle type $i$, $\nu_i$ the Fermi momentum and is linked to the number density

$$n_i = \langle \bar{\psi}_i \gamma^0 \psi_i \rangle = \frac{g_i \nu_i^3}{6\pi^2}. \tag{5}$$

Note that the masses of leptons take constant values with $m_e = 0.511$ MeV and $m_\mu = 105.66$ MeV. The chemical potentials $\mu_i$ and pressure $P$ can then be obtained according to the basic relations of standard thermodynamics.

For $ud$QM without leptons, to investigate its saturation properties, we then expand the energy per baryon to the second order [26], i.e.,

$$\varepsilon_{2f}(n_b, f_Z) = \varepsilon_0 + \frac{K_0}{18}\left(\frac{n_b}{n_0} - 1\right)^2 + \varepsilon_s(2f_Z - 1)^2, \tag{6}$$

where $n_b = (n_u + n_d)/3$ is the baryon number density, $f_Z = (2n_u - n_d)/(n_u + n_d)$ the charge fraction, $\varepsilon_0$ the minimum energy per baryon at saturation density $n_0$ and charge fraction $f_Z = 1/2$, $K_0$ the incompressibility parameter, and $\varepsilon_s$ the symmetry energy. The corresponding values obtained with equivparticle model using various combinations of parameters are then indicated in Table 1, where the symmetry energy is found to be increasing with $C_I$.

**Table 1.** The adopted parameter sets for the quark mass scaling in Equation (3) and the corresponding properties of $ud$QM.

| $C$ | $\sqrt{D}$ MeV | $C_I$ MeV/fm$^3$ | $n_0$ fm$^{-3}$ | $\varepsilon_0$ MeV | $K_0$ MeV | $\varepsilon_s$ MeV | $\sigma_{min}$ MeV/fm$^2$ | $\sigma_{crit}$ MeV/fm$^2$ |
|---|---|---|---|---|---|---|---|---|
| | | 0 | | | | 16.0 | 14.0 | 5.03 |
| $-0.5$ | 176 | 40 | 0.275 | 900.9 | 2571.5 | 36.5 | 13.1 | 14.5 |
| | | 80 | | | | 57.5 | 12.8 | 24.2 |
| | | 0 | | | | 15.0 | 12.4 | 4.24 |
| $-0.3$ | 167 | 40 | 0.241 | 902.0 | 2306.4 | 34.1 | 11.7 | 12.1 |
| | | 80 | | | | 53.5 | 11.4 | 20.1 |
| | | 0 | | | | 12.6 | 12.1 | 2.77 |
| 0.1 | 149 | 40 | 0.172 | 897.4 | 1942.0 | 27.8 | 11.6 | 7.58 |
| | | 80 | | | | 43.1 | 11.4 | 12.5 |
| | | 0 | | | | 10.4 | 7.40 | 1.77 |
| 0.5 | 135 | 40 | 0.120 | 904.6 | 1786.3 | 22.0 | 7.03 | 4.65 |
| | | 80 | | | | 33.5 | 6.87 | 7.65 |

## 3. $ud$QM Nuggets

At fixed surface tension values, the energy per baryon of $ud$QM nuggets can be obtained with a liquid-drop type formula, i.e.,

$$\frac{M}{A} = \varepsilon_{2f}(n_0, f_Z) + \frac{1}{5}\sqrt[3]{36\pi n_0}\alpha f_Z^2 A^{2/3} + \sigma\left(\frac{An_0^2}{36\pi}\right)^{-1/3}, \qquad (7)$$

where the bulk (first) term is fixed by Equation (6). The second term represents the Coulomb energy with $\alpha$ being the fine structure constant and the third term corresponds to the surface energy. By minimizing Equation (7) with respect to $f_Z$, one obtains the charge fraction for $\beta$-stable $ud$QM nuggets, i.e.,

$$f_Z = \frac{10\varepsilon_s}{\sqrt[3]{36\pi n_0}\alpha A^{2/3} + 20\varepsilon_s}. \qquad (8)$$

Then we can constrain the parameters by comparing the binding energy of $ud$QM nuggets with finite nuclei, where $ud$QM nuggets should always be unstable at $A \lesssim 300$ and stable as $A \to \infty$. At this moment, the heaviest $\beta$-stable nucleus is $^{266}$Hs with its energy per baryon being 931.74 MeV [75–77]. We thus require the energy per baryon obtained with Equation (7) at $A = 266$ should always be larger than that of $^{266}$Hs, where a lower limit for the surface tension value $\sigma_{min}$ is found for various combinations of parameters. The corresponding constraints are then indicated in Figure 1 for $C_I = 0$. The red curve indicates the boundary with the minimum energy per baryon $\varepsilon_0 = 922$ MeV, so that $ud$QM can be more stable than nuclear matter at $\delta = 0$ in the lower left region. If we demand $ud$QM star matter to be more stable than neutron star matter, the stable-unstable boundary is expected to shift slightly to the lower left region. Meanwhile, the surface tension value should be larger than $\sigma_{min}$, which favors the upper right region at a fixed $\sigma$. The combination of both constraints indicate an stability window for $ud$QM at a given surface tension value $\sigma$.

Finally, it is worth mentioning that the constraints in Figure 1 are insensitive to $C_I$, where we have only presented the results at $C_I = 0$.

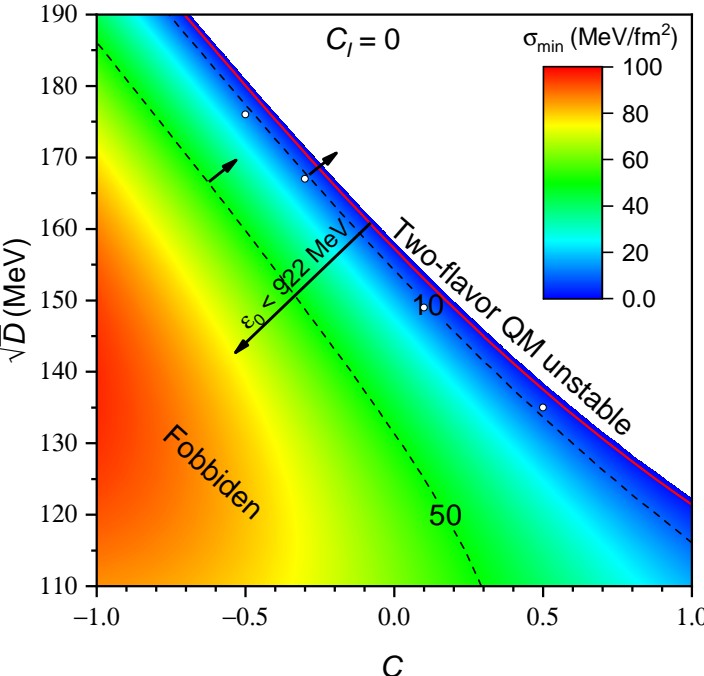

**Figure 1.** Various constrains for the parameters of the mass scaling in Equation (3) and minimum surface tension value $\sigma_{\min}$ for $ud$QM nuggets to be unstable comparing with $^{266}$Hs. The open circles correspond to the choices of parameters in Table 1.

The liquid-drop type formula in Equation (7) is valid for small $ud$QM nuggets. If we want to investigate the properties of $ud$QM nuggets at larger baryon numbers, the effects of charge screening and electrons should be considered [78]. In such cases, we adopt Thomas–Fermi approximation as was done in References [26,58–61]. The total energy of the system is obtained with

$$M = \int_0^\infty \left[ 4\pi r^2 E(r) + \frac{r^2}{2\alpha} \left( \frac{\mathrm{d}\varphi}{\mathrm{d}r} \right)^2 \right] \mathrm{d}r + 4\pi R^2 \sigma, \tag{9}$$

where the energy density $E$ is given by Equation (4) and the electric potential $\varphi$ by solving

$$r^2 \frac{\mathrm{d}^2\varphi}{\mathrm{d}r^2} + 2r \frac{\mathrm{d}\varphi}{\mathrm{d}r} + 4\pi\alpha r^2 \left( \frac{2}{3} n_u - \frac{1}{3} n_d - n_e \right) = 0. \tag{10}$$

The most stable structure of an $ud$QM nugget is fixed by minimizing Equation (9), which follows the constancy of chemical potentials, i.e.,

$$\mu_i(\vec{r}) = \sqrt{\nu_i(\vec{r})^2 + m_i(\vec{r})^2} + V_i(\vec{r}) = \text{constant}. \tag{11}$$

Here the vector potential is given by

$$V_i = \frac{\mathrm{d}m_{\mathrm{I}}}{\mathrm{d}n_i} \sum_{i=u,d} n_i^{\mathrm{s}} + q_i \varphi, \tag{12}$$

where the scalar density is

$$n_i^{\mathrm{s}} = \langle \bar{\psi}_i \psi_i \rangle = \frac{g_i m_i^3}{4\pi^2} g \left( \frac{\nu_i}{m_i} \right) \tag{13}$$

with $g(x) = x\sqrt{x^2 + 1} - \text{arcsh}(x)$. The quantities $n_i^{\text{s}}$, $n_i$, $\nu_i$, $m_i$ and $E$ represent the local properties of $ud$QM and vary with the space coordinates. Note that a density derivative term is added to the vector potential in Equation (12), which is essential in order to maintain the self-consistency of thermodynamics [62,68,79–82]. Meanwhile, since in Equation (3) we have added the third term to account for the symmetry energy of quark matter, in contrast to our previous findings, the density derivative term takes different forms for $u$ and $d$ quarks, i.e.,

$$\frac{\text{d}m_{\text{I}}}{\text{d}n_u} = \frac{1}{3}\frac{\text{d}m_{\text{I}}}{\text{d}n_{\text{b}}} + \frac{\text{d}m_{\text{I}}}{\text{d}\delta}\frac{3 - \delta}{3n_{\text{b}}}, \tag{14}$$

$$\frac{\text{d}m_{\text{I}}}{\text{d}n_d} = \frac{1}{3}\frac{\text{d}m_{\text{I}}}{\text{d}n_{\text{b}}} - \frac{\text{d}m_{\text{I}}}{\text{d}\delta}\frac{3 + \delta}{3n_{\text{b}}}. \tag{15}$$

In such cases, even though we have assumed the same equivalent masses for $u$ and $d$ quarks, their vector potentials take different forms, i.e., contributions to the isovector-vector channel.

Beside the constancy of chemical potentials in Equation (11), the dynamic stability of the quark-vacuum interface should also be fulfilled, which determines the radius of the quark core $R$ with

$$P_{\text{QM}}(R) = \frac{2\sigma}{R}. \tag{16}$$

The obtained energy per baryon for $ud$QM nuggets are then presented in Figure 2, where we have adopted the parameters indicated in Table 1 as well as the smallest possible surface tension value $\sigma = \sigma_{\text{min}}$. It is found that the energy per baryon for various cases crosses at $A = 266$ and is generally decreasing with $A$. However, if large symmetry energies with $C_I = 40$ and 80 MeV/fm$^3$ were considered, large $ud$QM nuggets are destabilized and there may exists $ud$QM nuggets at $A \approx 1000$ that are more stable than others. Particularly, there are even cases where the minimum energy per baryon of $ud$QM star matter is larger than 930 MeV but that of $ud$QM nuggets smaller than 930 MeV, e.g., $C = -0.5$, $\sqrt{D} = 176$ MeV, and $C_I = 40$ MeV/fm$^3$. Such kind of scenarios occur when the surface tension value is smaller than the critical one $\sigma_{\text{crit}}$, which can be roughly estimated with Equation (1) with the corresponding values presented in Table 1. In such cases, the surface of an $ud$QM star will fragment into a lattice of $ud$QM nuggets immersed in a sea of electrons, similar to the cases of strange stars with crusts of strangelets [51,52].

The charge-to-mass ratios of $ud$QM nuggets are indicated in Figure 3, which are decreasing with baryon number due to Coulomb repulsion. Additionally, for the cases with a larger symmetry energy (larger $C_I$), $ud$QM nuggets are more positively charged, which gives larger Coulomb energy and leads to a barrier of energy per baryon as we increase $A$ for the cases with $\sigma < \sigma_{\text{crit}}$.

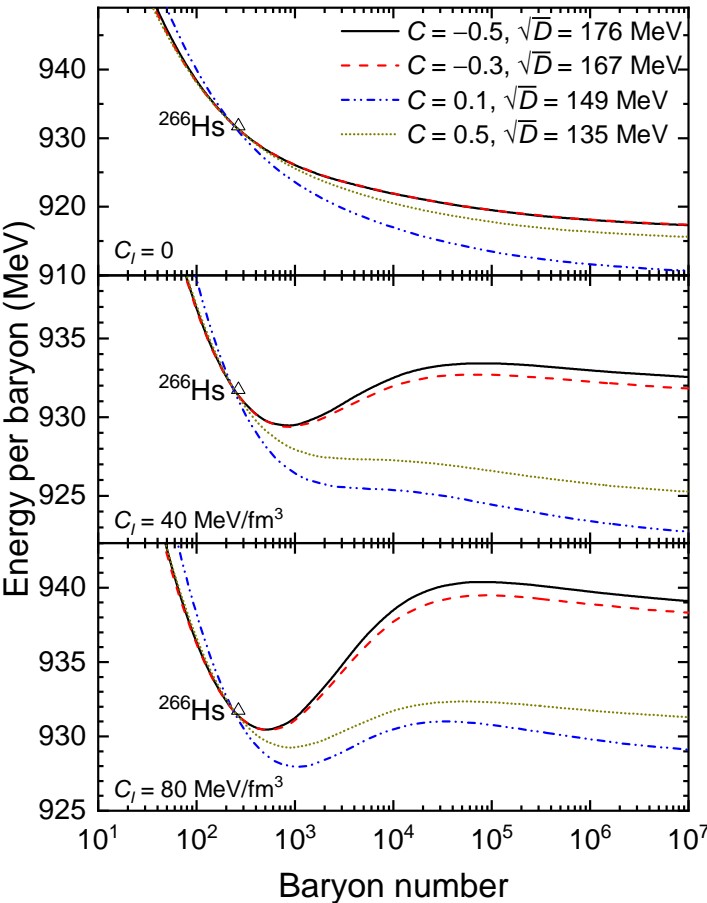

**Figure 2.** Energy perbaryon for *ud*QM nuggets as functions of the baryon number *A*. The experimental data for the heaviest *β*-stable nuclei $^{266}$Hs is obtained from the 2016 Atomic Mass Evaluation [75–77].

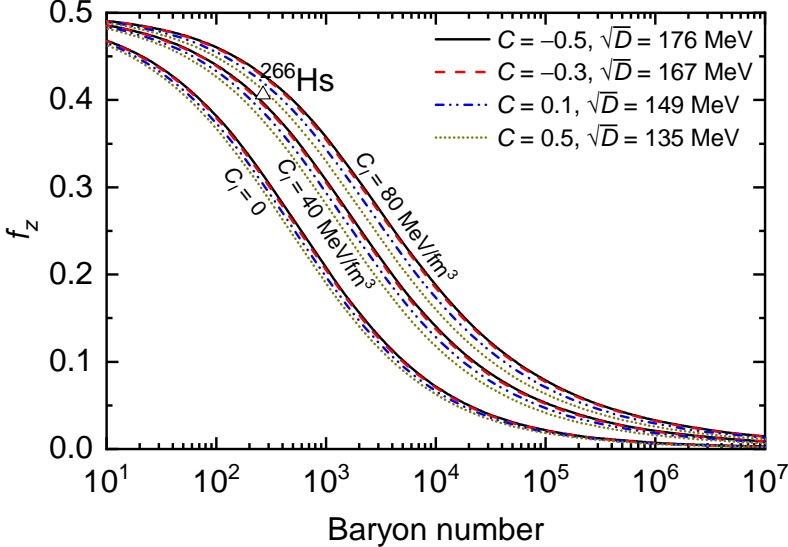

**Figure 3.** Same as Figure 2 but for the charge-to-mass ratio of *ud*QM nuggets.

## 4. *ud*QM Star

By fulfilling simultaneously the local charge neutrality condition $\frac{2}{3}n_u - \frac{1}{3}n_d = n_e + n_\mu$ and *β*-stability condition $\mu_u + \mu_e = \mu_u + \mu_\mu = \mu_d$, the equation of state (EOS) for *ud*QM

star matter can be fixed based on the equivparticle model illustrated in Section 2. Then the structure of *ud*QM star can be obtained by solving the TOV equation

$$\frac{dP}{dr} = -\frac{GME}{r^2}\frac{(1+P/E)(1+4\pi r^3 P/M)}{1-2GM/r}, \tag{17}$$

$$\frac{dM}{dr} = 4\pi Er^2, \tag{18}$$

where the gravity constant is taken as $G = 6.707 \times 10^{-45}$ MeV$^{-2}$. The tidal deformability can also be estimated with perturbation method, where a special boundary treatment on the surface is required for bare *ud*QM star [83–86].

The energy per baryon of *ud*QM nuggets is decreasing monotonously with baryon number for certain choices of parameters in Figure 2, which excludes a crust of *ud*QM nuggets. The corresponding mass-radius relations of bare *ud*QM stars are then presented in Figure 4. Note that the third term in the quark mass scaling in Equation (3) generally introduces an repulsive interaction among quarks, so that larger masses and radii are obtained for larger symmetry energy (larger $C_I$). The obtained maximum masses for the parameter sets ($C$, $\sqrt{D}$ in MeV): ($-0.5$, 176), ($-0.3$, 167), (0.1, 149), and (0.5, 135) reach the observational mass ($2.14^{+0.20}_{-0.18}M_\odot$, 95.4% credibility) of PSR J0740+6620 [87], while the radii of the two-solar-mass stars are consistent with that of PSR J0740+6620 ($12.39^{+1.30}_{-0.98}$ km and $2.072^{+0.067}_{-0.066}M_\odot$) only for $C = 0.1$ and $\sqrt{D} = 149$ MeV [88]. Nevertheless, if we examine the tidal deformation of those stars, only the *ud*QM stars obtained with the parameter sets ($-0.5$, 176) and ($-0.3$, 167) meet the constraints $70 \leq \Lambda_{1.4} \leq 580$ from the binary neutron star merger event GW170817 [89].

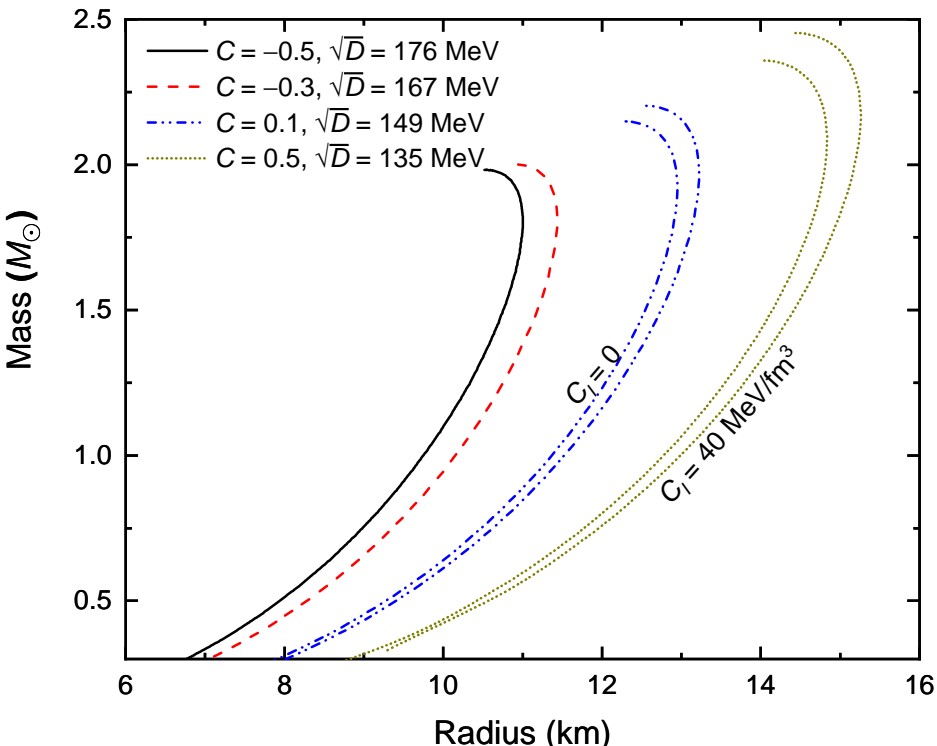

**Figure 4.** Mass-radius relation of *ud*QM stars without crusts, where $C_I = 0$. For the two cases with larger radii, $C_I = 40$ MeV/fm$^3$ are adopted.

For the cases where *ud*QM nuggets at certain sizes ($A \approx 1000$) are more stable than others, the surface of *ud*QM star will fragment into a crystalline crust. To obtain the EOSs of the *ud*QM star crust, as an rough estimation, we neglect the effects of charge screening and assume vanishing surface tension, i.e., Gibbs construction. By equating the pressures

of the quark phase $[P_{QM}(\mu_b, \mu_e) + P_e(\mu_e)]$ and electrons $[P_e(\mu_e)]$, we find that the pressure of pure quark matter should vanish, i.e.,

$$P_{QM}(\mu_b, \mu_e) = 0. \tag{19}$$

Then the pressure of the nonuniform phase is exactly the pressure of electrons $P = P_e(\mu_e)$. The volume fraction $\chi$ of the quark phase is fixed according to the global charge neutrality condition, i.e.,

$$\chi \left( \frac{2}{3} n_u - \frac{1}{3} n_d \right) = n_e. \tag{20}$$

Combining both the quark phase and electron phase, the energy density is determined by

$$E = \chi E_{QM}(\mu_b, \mu_e) + E_e(\mu_e). \tag{21}$$

The obtained EOSs are presented in Figure 5, where the nonuniform phase takes place at $E \lesssim 200$ MeV/fm³. Note that the nonuniform phase in $ud$QM star is similar to that of the outer crust of neutron stars. Since the surface tension is nonzero and the energy per baryon of the most stable $ud$QM nugget does not reach $\varepsilon_0$, we expect the formation of various geometrical structures in $ud$QM stars. A detailed investigation is thus necessary to obtain the realistic structures and EOSs [50,52], which is intended in our future works. Additionally, we should mention that the possible strong attractive interactions among quark clusters could result in very interesting conclusions [90–92], where the formations of strangeon matter and strangeon stars are expected [93,94].

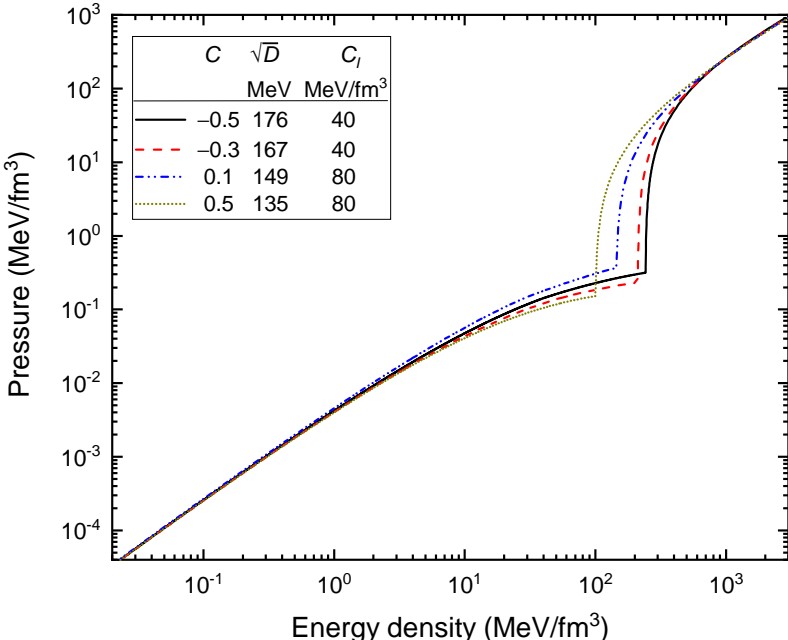

**Figure 5.** Equation of state for $ud$QM star matter, which includes both the uniform phase at large densities and nonuniform phase at small densities.

Based on the EOSs presented in Figure 5, we solve the TOV Equation (17) and obtain the structures of $ud$QM stars with crusts, where the mass-radius relations are presented in Figure 6. Due to the presence of a crust, the mass-radius relations are similar to those of neutron stars, where the radius is increasing as we decrease the density in the center. This is essentially different from that of hybrid stars with unstable quark matter, where the deconfinement phase transition would reduce the radius and even lead to high-mass twins in case of a strong first-order phase transition [95]. To show this explicitly, the

mass-radius relation for hybrid stars is presented in Figure 5, which is obtained with the combination of a density functional PKDD for nuclear matter, $p$QCD with $C_1 = 2.5$ and $\Delta\mu = 770$ MeV for quark matter, and a surface tension value $\sigma = 5$ MeV/fm$^2$ as indicated in Reference [96]. For $ud$QM stars, it is found that the maximum masses for the parameter sets ($C$, $\sqrt{D}$ in MeV): $(-0.3, 167)$, $(0.1, 149)$, and $(0.5, 135)$ reach the observational mass $(2.14^{+0.20}_{-0.18} M_\odot$, 95.4% credibility) of PSR J0740+6620 [87], while the radii of the two-solar-mass stars coincide with that of PSR J0740+6620 ($12.39^{1.30}_{-0.98}$ km and $2.072^{+0.067}_{-0.066} M_\odot$) only for $C = -0.3$ and $\sqrt{D} = 167$ MeV. Similar as the cases of bare $ud$QM stars, the GW170817 constraint $70 \leq \Lambda_{1.4} \leq 580$ is consistent with the parameter sets $(-0.5, 176)$ and $(-0.3, 167)$ [89]. Combined with all these constraints, only the case of $C = -0.3$ and $\sqrt{D} = 167$ MeV is consistent with various pulsar observations.

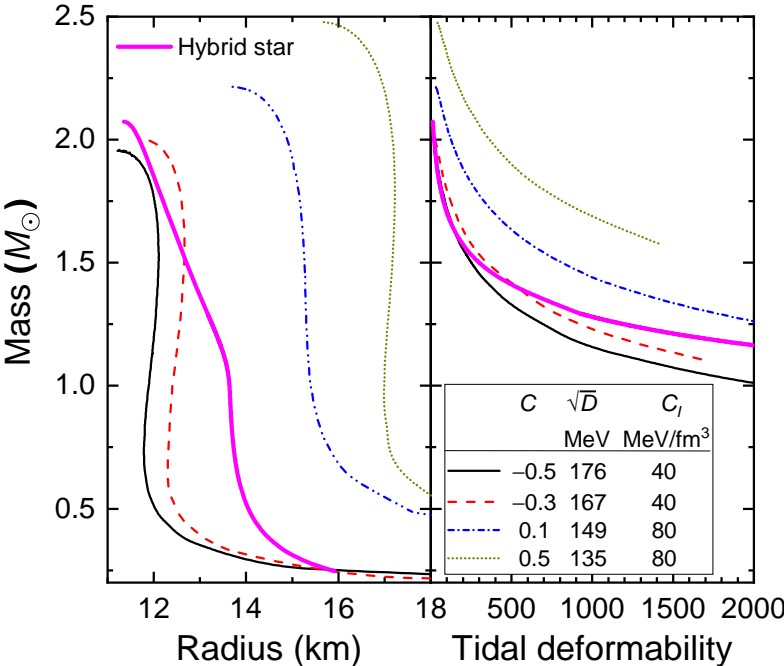

**Figure 6.** Mass-radius relations (left) and tidal deformabilities (right) of $ud$QM stars with crusts, where the EOSs presented in Figure 5 were adopted. The corresponding values for typical hybrid stars from Reference [96] are presented as well.

In addition to the pulsar-like objects discussed so far, it was realized that for SQM there exists low-mass large-radius strangelet dwarfs if the surface tension is small enough [97]. A strangelet dwarf is comprised of a charge separated phase, which is energetically favorable to form crystalline structures with strangelets and electrons. Adopting the EOSs with $P_{\text{QM}} = 0$ in Figure 5 and solving the TOV Equation (17), we have observed similar objects comprised of only $ud$QM nuggets and electrons as indicated in Figure 7. At sufficiently low temperatures, the $ud$QM nuggets and electrons form crystalline structures inside the star, which is in analogy with white dwarfs comprised of nuclei and electrons. We thus refer to them as $ud$QM dwarfs. Comparing with strangelet dwarfs [97], the masses of $ud$QM dwarfs are much larger and approaching to the Chandrasekhar limit ($\sim$1.4 $M_\odot$), which is mainly due to the large charge-to-mass ratio of $ud$QM nuggets. In such cases, it is likely that some of the observed white dwarfs may in fact be $ud$QM dwarfs. Nevertheless, it is worth mentioning that the results indicated in Figure 7 should be considered as an upper limit, where the emergence of geometrical structures with various surface tension values is expected to play an important role [97]. A detailed investigation is thus necessary and intended for our future study.

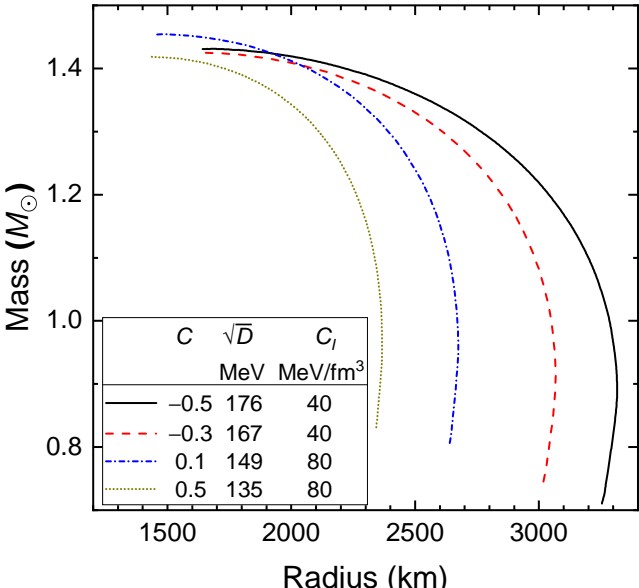

**Figure 7.** Same as Figure 6 but for the mass-radius relations of *ud*QM dwarfs.

## 5. Conclusions

In this work we investigate the stability of *ud*QM and the properties of *ud*QM nuggets, *ud*QM stars, and white dwarfs comprised of *ud*QM nuggets. The properties of *ud*QM are obtained based on equivparticle model [62,63,68], where an additional term is introduced to account for the quark matter symmetry energy. By comparing the binding energies of *ud*QM nuggets and the heaviest $\beta$-stable nucleus $^{266}$Hs, the stability window for *ud*QM is obtained, where there exists a minimum surface tension value below which finite nuclei cannot decay into *ud*QM nuggets. The properties of *ud*QM nuggets are then examined considering the effects of charge screening and electrons [78]. By adopting the minimum possible surface tension values, there are *ud*QM nuggets at certain size that are more stable than others if a large symmetry energy is adopted. In such cases, a crust made of *ud*QM nuggets is expected in *ud*QM stars, where we have obtained the EOSs adopting Gibbs construction. It is found that the mass-radius relation of *ud*QM stars resembles that of traditional neutron stars. Meanwhile, similar to the cases of strange stars [97], a new family of white dwarfs comprised entirely of *ud*QM nuggets and electrons is obtained, where its maximum mass approaches to the Chandrasekhar limit due to the large charge-to-mass ratio of *ud*QM nuggets.

**Author Contributions:** Conceptualization, C.-J.X., J.-F.X., G.-X.P. and R.-X.X.; Data curation, C.-J.X.; Funding acquisition, C.-J.X.; Investigation, J.-F.X.; Methodology, G.-X.P. and R.-X.X.; Writing—original draft preparation, L.W. and J.H.; Writing—review and editing, J.-F.X., G.-X.P. and R.-X.X. All authors have read and agreed to the published version of the manuscript.

**Funding:** This research was funded by by National SKA Program of China No. 2020SKA0120300, National Natural Science Foundation of China (Grant Nos. 11705163, 12005005, 11947098, 11875052, and 11673002), Ningbo Natural Science Foundation (Grant Nos. 2019A610066 and 2019A610092), and key research projects of universities in Henan province (Grant No. 20A140003).

**Institutional Review Board Statement:** Not applicable.

**Conflicts of Interest:** The authors declare no conflict of interest.

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
