# Peer review of "Stable Up-Down Quark Matter Nuggets, Quark Star Crusts, and a New Family of White Dwarfs"

_galaxies, doi:10.3390/galaxies9040070_

Round 1

Reviewer 1 Report

The work is very significant with interesting results, the only that I would like to see is an expansion regarding the white dwarfs' family, maybe separating it from fig 6 and having a more detailed explanation. The exploration seems shy (since it appears in the tittle, I would be expecting the subject to have more protagonism. The hypothesis of this new white dwarfs maybe can be important for type Ia supernovae.

Author Response

Dear Referee,

Thank you very much for your careful review and kind recommendation! Your comments and suggestions are very helpful to increase the quality of our manuscript, and we have made modifications based on them. The modifications are listed as follows:

  1. The right panel of Fig. 6 is now separated as Fig. 7 in the modified manuscript, where the mass-radius relations of udQM dwarfs are presented.
  2. More discussions are added to the last paragraph of Sec. 4, which becomes:

In addition to the pulsar-like objects discussed so far, it was realized that for SQM there exists low-mass large-radius strangelet dwarfs if the surface tension is small enough [97]. A strangelet dwarf is comprised of a charge separated phase, which is energetically favorable and forms crystalline structures with strangelets and electrons. Adopting the EOSs with PQM = 0 in Fig. 5 and solving the TOV equation (17), we have observed similar objects comprised of only udQM nuggets and electrons as  indicated in Fig. 7. At sufficiently low temperatures, the udQM nuggets and electrons form crystalline structures inside the star, which is in analogy with white dwarfs comprised of nuclei and electrons. We thus refer to them as udQM dwarfs. Comparing with strangelet dwarfs [97], the masses of udQM dwarfs are much larger and approaching to the Chandrasekhar limit (~1.4 M_sun), which is mainly due to the large charge-to-mass ratio of udQM nuggets. In such cases, it is likely that some of the observed white dwarfs may in fact be udQM dwarfs. Nevertheless, it is worth mentioning that the results indicated  in Fig. 7 should be considered as an upper limit, where the emergence of geometrical structures with various surface tension values are expected to  play important roles [97]. A detailed investigation is thus necessary and intended for our future study.

Yours sincerely,

Cheng-Jun Xia on behalf of the authors

Reviewer 2 Report

The paper studies ud-quark matter which has binding energy which is larger than that of nuclear matter. This means that the nuclear matter is in metastable state. The idea is a copy of the original Bodmer-Witten hypothesis with the only difference that the strange quark is neglected. Because of the uncertainty of the parameters and physics in general, the authors find the parameter space that conforms the idea of ud quark matter. The paper is fine . It would be useful, in my opinion, to compare the MR-diagrams of ud quark matter to that of hybrid stars, see for example, arXiv:1709.08857

Author Response

Dear Referee,

Thank you very much for your careful review and kind recommendation! Your comments and suggestions are very helpful to increase the quality of our manuscript, and we have made modifications based on them. The modifications are listed as follows:

  1. The mass-radius relation of typical hybrid stars has now added to Fig. 6, which is distinctively different from that of udQM stars.
  2. More discussions are added following the second sentence of the penultimate paragraph in Sec. 4, i.e.,

This is essentially different from that of hybrid stars with unstable quark matter, where the deconfinement phase transition would reduce the radius and even lead to high-mass twins in case of a strong first-order phase transition [95]. To show this explicitly, the mass-radius relation for hybrid stars is presented in Fig. 5, which is obtained with the combination of density functional PKDD for nuclear matter,pQCD with C_1=2.5 and ∆μ=770MeV for quark matter, and a surface tension value σ = 5 MeV/fm^2as indicated in Ref. [96].

Yours sincerely,

Cheng-Jun Xia on behalf of the authors